# A LLM-Refined Dynamic Topic Clustering Framework for Business Forecasts with Large Corpora

## Abstract

We propose a new framework for topic modeling of financial news, integrating large language models (LLMs) with document embeddings, clustering, and GPT-4-based topic refinement. Our method filters and processes Dow Jones financial articles, embeds them using OpenAI models, applies HDBSCAN+UMAP for clustering, and uses GPT-4 to generate, deduplicate and refine topic descriptions. Our model generates higher quality and more stable topics than conventional topic models such as LDA and BERTopic, and substantially improves forecasting accuracy. The generated topics are highly interpretable and distinct, contain rich information for the state of economy, and have high predictive power for macroeconomic and stock market performance. This study represents the first application of GPT-4-assisted clustering refinement to topic modeling and financial forecasts.

## 1 Introduction

The vast volume of financial news compiled daily by news media contains rich information on market events, corporate development, and the state of the economy. Effectively summarizing this textual data into structured signals is crucial for tasks such as forecasts of business cycles and market performance. Topic modeling offers a way to retrieve latent themes in text corpora without supervision. In the finance domain, topics derived from news can capture emergent risks or trends in real time that conventional structured economic indicators based on historical data may miss. Prior research has shown that incorporating news-derived topics or textual factors can improve forecasts of economic activities and asset returns. For instance, Ellingsen et al. (2020) use 22.5 million Dow Jones news articles to extract information signals and find that the news-based features significantly enhance predictions of consumption growth beyond what traditional time-series models can achieve. This motivates the use of text data in financial news press to uncover latent informative themes for improving financial forecasts and economic models. Conventional finance research has relied on rudimentary methods to extract signals from textual data, such as dictionary-based text classification or the bag-of-words (BoW) model. Only recently have finance researchers begun to use Latent Dirichlet Allocation (LDA) to generate economic topics from news articles by extracting conceptual representations of text (see Bybee et al. (2024)). However, LDA depends solely on term-frequency patterns, offering little control over the specificity or interpretability of its outputs. Because it ignores context, LDA often incurs significant information loss, producing imprecise or ambiguous topics. Topics are represented as unordered lists of probable terms, which frequently include incoherent or generic words, leaving analysts to "read the tea leaves" to infer meaning. In financial applications, LDA may group unrelated texts under the same topic or fragment a coherent theme across multiple topics. Moreover, the model requires prespecifying the number of topics ($K$), and poor choices can yield redundant or overly broad clusters. These limitations reduce the value of LDA-based models for high-stakes financial analysis, where interpretability and well-defined topics are critical.

Recent advances in large language models (LLMs) offer promising alternatives. State-of-the-art models such as GPT demonstrate strong abilities in understanding and summarizing domain-specific text. Notably, ChatGPT has shown a high degree of financial literacy in interpreting news Chiu et al. (2025), suggesting that LLMs can serve as powerful engines for topic discovery in finance research.

In this paper, we introduce a novel two-stage LLM-driven topic analysis framework for financial news, which we call **FinTOA** (Financial Topic Attention). The goal of FinTOA is to transform the

vast flow of unstructured news into structured and interpretable time-series signals that can be directly used for forecasting financial markets and macroeconomic activity. In the first stage, FinTOA extracts a set of high-quality, distinct, and human-readable topics from a large corpus of financial news. In the second stage, it quantifies each article's relevance to these topics by prompting an LLM to assign "attention" scores across the discovered topics. By aggregating these article-level attention scores over time (e.g., daily or monthly), FinTOA produces structured topic-attention signals that bridge textual narratives with financial and economic outcomes. This integrated framework combines interpretability(through concise and meaningful topic labels) with predictive power, as the attention signals can be systematically linked to subsequent market returns and macro indicators.

Our framework builds on and extends the recent Reuter et al. (2024) model (TopicGPT), a prompt-based topic modeling approach that integrates GPT's strengths into the topic extraction process. TopicGPT demonstrates how large language models can generate natural-language topic descriptions, avoiding the incoherent or overly generic word lists that arise in traditional methods such as LDA.[1] Moreover, TopicGPT supports interactive refinement (e.g., merging overlapping topics) through follow-up prompts, yielding human-centric, interpretable topics that remain grounded in the underlying text data. FinTOA leverages these advantages in the finance domain and extends them by adding a second-stage attention mechanism that links each article to the discovered topics. This novel extension transforms static topic descriptions into dynamic time-series signals, enabling real-time forecasting and richer interpretability in financial applications.

This paper makes three major contributions to the literature. First, we uncover meaningful topics in financial news. To our knowledge, our paper represents the first attempt to use state-of-the-art Large Language Models in topical analysis in financial research. We address domain-specific preprocessing challenges to ensure that the LLM input is representative and noise-free. This procedure results in high-quality topics that are coherent and domain-specific.

Second, we propose a LLM prompting technique to measure each article's attention to all the identified topics. Instead of relying on probabilistic topic mixtures as in LDA, we use GPT-4 to assign relevance scores to indicate how strongly an article is related to each topic. We then aggregate these LLM-based relevance scores into daily and monthly topic attention time series. These signals quantify the prevalence of each topic over time, which are then fed into the predictive model.

Third, we demonstrate that the LLM-generated topics and their attention intensities are both qualitatively and quantitatively effective. Qualitatively, the topics are more coherent, specific, and distinctive than those generated by the LDA model, making them immediately interpretable with high precision. Quantitatively, when used as features in the predictive model, these topic-based signals significantly improve the performance of stock return and macroeconomic forecasts, substantially outperforming conventional methods like the LDA model or static keyword-based topic labeling. We provide visualizations of the learned topics and their temporal dynamics, illustrating that our FinTOA approach produces intuitive topic groupings and insights that can be readily understood by analysts.

The remainder of this paper is organized as follows. Section 2 describes our methodology, including data preprocessing, the LLM-based topic extraction, and the construction of topic attention features. Section 3 presents experiments to evaluate the quality of the derived topics and the predictive value of the extracted signals, with results highlighting the distinct advantages of our approach. Section 4 offers another experiment of the topics and their application on stock market returns. Finally, Section 5 concludes the paper with implications of our findings and future directions.

## 2 METHODOLOGY

Our methodology consists of four main components, which together form the basis for topic discovery and utilization of financial news. The implementation of this framework can be summarized as follows.

Data Preprocessing: We combine headlines and body text from news articles and select only full-length articles for further processing.

---

[1]Specifically, topics from LDA are typically represented as bags of words—lists of the most probable terms—that often include incoherent or generic words.

Topic Extraction via GPT-based Modeling: We embed documents, perform clustering to identify topics, and use a large language model (LLM) to generate descriptive labels for each topic.

Topic Deduplication and Refinement: We merge semantically overlapping topics based on LLM feedback to ensure each final topic is distinct and meaningful.

Feature Construction for Prediction: We transform the stream of topic-tagged articles into time-series features (e.g., daily topic frequencies) for downstream predictive tasks (detailed later in the experiment section).

## 2.1 DATA SOURCES AND PREPROCESSING

We process a comprehensive Dow Jones news dataset that spans 1979-2023. Figure 1 shows the monthly word and article counts for this dataset. To prepare the data for topic modeling, we apply the following preprocessing steps:

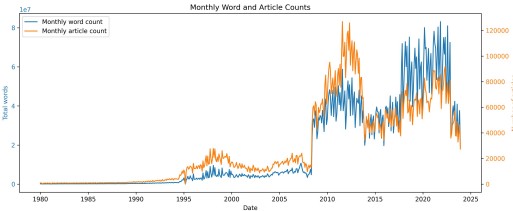

Figure 1: Monthly word and article count for Dow Jones news dataset.

First, we construct a single combined document string by concatenating the headline and the body. We next filter out pieces that are too short or not substantive enough for reliable topic modeling. The Dow Jones news stream contains many brief bulletins that lack context of substance. We implement a heuristic filter that checks for signs of a full-length article. In our implementation, an item passes the filter if (a) the headline is at least 20 characters, and (b) the body contains at least three paragraphs of a over 50 characters. In our dataset, roughly 60 percent of the raw news items passed this full-article filter. The selected articles in this step form the corpus used for topic modeling.

## 2.2 TOPIC EXTRACTION WITH GPT-BASED MODELING

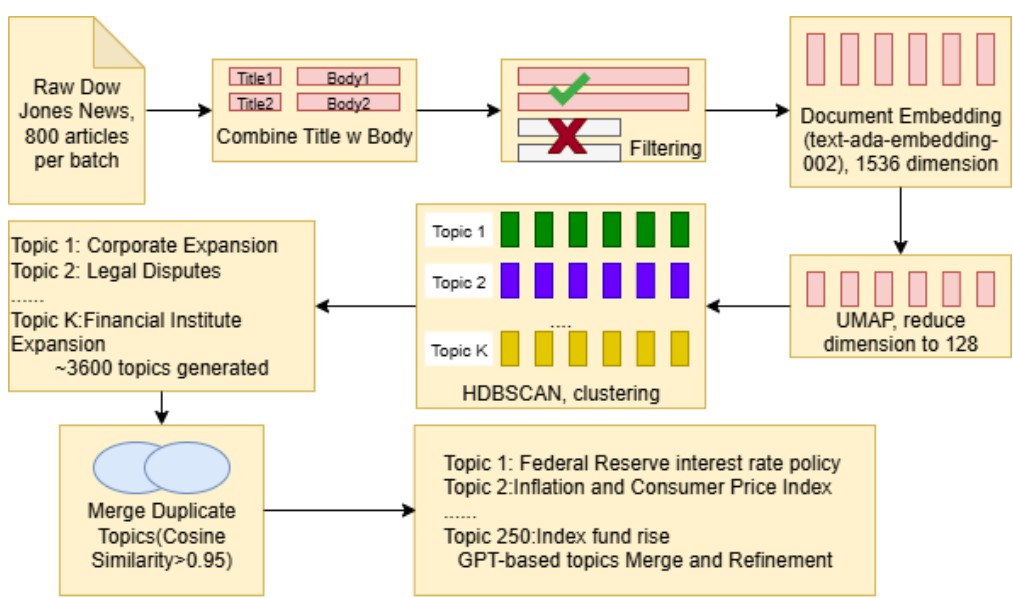

Figure 2: Overview of the FinTOA framework. The process comprises (a) topic extraction using document embeddings and GPT-4 labeling, (b) deduplication via similarity metrics and LLM refinement

To uncover the latent thematic structure in our large-scale financial news corpus, we developed a pipeline for fully unsupervised topic discovery. Rather than embedding and clustering all $\sim 1.2$ million articles at once, we split the 1979–2023 corpus into batches of roughly 800 articles each.

For each batch, we first compute high-dimensional document embeddings using OpenAI's latest models. Because these embeddings live in a space of over a thousand dimensions, which is too complex to cluster directly, we apply UMAP (Uniform Manifold Approximation and Projection) to reduce the dimension. UMAP is a nonlinear dimensionality-reduction technique that "unfolds" the high-dimensional data into 128-dimensional representation, carefully preserving the notion of which articles are nearest neighbors (so that articles on the same theme stay close together) while maintaining the global layout of clusters.

Next, we feed the UMAP output into HDBSCAN (Hierarchical Density-Based Spatial Clustering of Applications with Noise). HDBSCAN identifies groups of articles by finding regions in the low-dimensional space where points (i.e. financial news) are densely packed, automatically determining both the number of clusters and which points are too isolated ("noise"). This means we do not need to guess ahead of time how many topics exist in each batch, and outlier articles simply remain unclustered rather than forcing them into a poor-fit group.

Finally, once each HDBSCAN cluster is formed, we prompt GPT-4 to generate a concise, human-readable label that captures the common theme of its member articles. Repeating this process across all batches yields an initial pool of over 3,600 candidate topics, each reflecting a distinct financial theme that may recur in different periods of our dataset.

## 2.3 Topic Deduplication and Refinement

Even with the careful design of the clustering and LLM labeling steps, the initial set of topics may contain redundancies or overly broad categories, especially since we merged the results from many batch runs. We perform a refinement step to address these potential issues.

First, we refine the embeddings generated to come up with more precise topics. This is achieved by computing the pairwise cosine similarity between the embeddings of all topic labels. If the cosine similarity between two topics exceeds a threshold of 0.95, we automatically merge these topics into a single representative topic. Next, We feed the entire list of initial topic labels and descriptions into a GPT-4 based review process. We instruct the LLM to examine the list and identify any topics that are semantically overlapping or duplicative, and suggest merges where appropriate. The instruction emphasizes that each final topic should be unique and non-overlapping, and that we want to preserve the breadth of coverage of financial themes.

After merging duplicates and dropping overly generic categories, we arrive at a refined set of topics, denoted $T_1', T'2, \ldots, T'K$ (in our experiments, the initial 3600 raw topics are consolidated down to 250 distinct topics). Each $T_i'$ now represents a unique thematic category in the financial news, with a clear descriptive label. This generates an enumerated list of well-defined financial news topics and a classification of the historical news articles into these viable topics.

## 2.4 Topic-Driven Feature Construction for Prediction

With the refined set of financial topics obtained in the preceding procedure, the next critical step is to quantify the level of attention each topic receives over time. Unlike traditional topic modeling methods that assign articles to single topics based on clustering or probability distributions, our approach exploits the generative capability of large language models to measure how closely each article is related to a specific topic in the list we identify.

Specifically, for each news article, we construct an LLM prompt designed to evaluate the relevance of an article to each of our established topics. The LLM, i.e., the GPT-4o-mini used in our implementation, is prompted with the article's content alongside a list of all final topics, and then asked to output an attention distribution across these topics. This yields a per-article topic attention vector $\mathbf{a}(n) \in \mathbb{R}^{K'}$, where each entry $a_i(n)$ indicates the attention that article $n$ places on topic $T_i'$. By leveraging GPT-4's nuanced semantic understanding, this attention distribution accurately reflects the multifaceted nature of financial news articles, which often sprawl multiple topics.

After obtaining these article-specific attention distributions, we aggregate them over daily and monthly windows to create structured numerical features for predictive modeling. Formally, for each day $d$, we aggregate the attention vectors of all articles published on that day: $\mathbf{x}(d) = \sum_n \mathbf{a}(n)/n$,

where $\mathbf{x}(d)$ is a daily aggregated topical attention vector representing the total news attention each topic pays to on day $d$ and n is all the articles on that day. Monthly attention vectors are formed by averaging daily vectors within the corresponding month.

These aggregated attention scores offer a nuanced and continuous measurement of how the thematic focus in financial news evolves over time. For example, an increase in daily attention scores for the topic *"Federal Reserve Interest Rate Policy"* may indicate heightened market sensitivity to the upcoming central bank announcement. By tracking these topic attention dynamics, our approach can capture forward-looking signals that are missed by simple article counts or traditional topic or term frequency measures.

The resulting daily or monthly time-series vectors $\mathbf{x}(d)$ can then be directly integrated as input features into conventional financial predictive models, such as logistic regression, ridge regression, or other machine learning approaches. In subsequent experimental evaluations, we demonstrate how incorporating these fine-grained attention-based topic features can significantly improve the forecasting efficiency across various financial prediction tasks, including stock market and macroeconomic performance forecasts.

In summary, our model capitalizes on GPT-4's exceptional semantic reasoning capability to translate raw financial text into rich, structured topic attention signals. This approach provides interpretable and timely insights into the shifting thematic landscape of financial news, ultimately enabling more accurate and actionable financial predictions.

## 3  EXPERIMENTS AND RESULTS

### 3.1  TOPIC QUALITY EVALUATION

After the model generates the output, we evaluate the quality of the discovered topics using the following four commonly used metrics (Isonuma et al. (2024)).

Topic Coherence: Measure the semantic coherence of each topic's top words using a coherence score $Cv$ (Röder et al. (2015)). Higher coherence indicates that the top words in a topic are more related to each other (e.g., appearing together in a reference corpus).

Topic Uniqueness ($TU$): Measure the diversity of topics by quantifying the overlap of top words between topics (Nan et al. (2019)). A $TU$ value closer to 1 means each topic's top words are largely distinct from those of other topics, indicating high diversity and minimal redundancy across topics.

Document Coverage ($DC$): Assess the extent to which the set of discovered topics covers the content of documents. We compute $DC$ as the fraction of documents that are "covered" by at least one topic. A higher DC means the topics collectively represent a larger portion of the corpus.

Factuality ($Fa$): Evaluate how grounded the topic descriptions are in the source text. Factuality is defined as the proportion of each topic's top words that appear in the original corpus. An $Fa$ of 1.0 indicates that all topic keywords are directly drawn from the documents. This metric is important for models, such as LLM-based topic models, that may generate words not present in the data.

Table 1 reports the results of evaluation using the four topic quality metrics for the benchmark models (LDA in Blei et al. (2003) and BERTopic in Grootendorst (2022)) versus our FinTOA model. As shown, overall, FinTOA achieves the highest performance based on these metrics. In particular, FinTOA's topics have substantially higher coherence than those generated by LDA and BERTopic. FinTOA also scores the highest in topic uniqueness ($TU = 0.90$). FinTOA balances topic diversity with strong document coverage, it covers about 95 percent of documents. Although LDA achieves slightly higher marginal coverage of documents, this comes at a large cost of less coherence, as evidenced by LDA's low $Cv$. Finally, all models score highly on factuality. Overall, the results of topic quality evaluation show that FinTOA produces more coherent, diverse, and representative topics than the alternative models, with solid factual grounding.

### 3.2  FORECASTING WITH TOPIC ATTENTION

Having established the quality of FinTOA's topics, we next investigate their usefulness for forecasting economic and financial activities. To directly compare with the performance of the Bybee et al. (2024) model that uses LDA to extract topics, we employ a similar experiment design that uses news-based topic attention to predict macroeconomic dynamics and financial market outcomes. In this setting, the topic attention time series refers to the monthly proportion of news content devoted

| Model | Coherence ($Cv$) | Uniqueness ($TU$) | Coverage ($DC$) | Factuality ($Fa$) |
|---|---|---|---|---|
| LDA Blei et al. (2003) | 0.45 | 0.80 | 0.90 | 1.00 |
| BERTopic Grootendorst (2022) | 0.50 | 0.85 | **0.96** | 1.00 |
| FinTOA(GPT-4o) | **0.57** | **0.90** | 0.95 | **1.00** |

Table 1: Topic quality metrics (higher is better) for three topic modeling methods: classical LDA, BERTopic, and the proposed FinTOA. FinTOA outperforms the baselines on Topic Coherence ($Cv$), Topic Uniqueness ($TU$), and maintaining perfect Document Coverage ($DC$) and perfect Factuality ($Fa$). Overall, the results suggest our FinTOA model produces more coherent and comprehensive topics for the financial news corpus.

| Industrial Prod. Growth | | | Employment Growth | | | Stock Returns | | | Market Volatility | | |
|---|---|---|---|---|---|---|---|---|---|---|---|
| *Selected Topic* | Coeff. | *p*-val | *Selected Topic* | Coeff. | *p*-val | *Selected Topic* | Coeff. | *p*-val | *Selected Topic* | Coeff. | *p*-val |
| Industrial production figures | +0.38 | 0.03 | Unemployment and job reports | -0.31 | 0.00 | Stock market performance | +0.25 | 0.05 | Options market and VIX | +0.40 | 0.01 |
| Wage growth | +0.22 | 0.01 | Labor market participation | +0.30 | 0.02 | Corporate earnings reports | +0.18 | 0.10 | Credit risk | +0.33 | 0.04 |
| Wage inequality | -0.20 | 0.05 | NASDAQ growth | -0.25 | 0.03 | Government shutdowns | -0.20 | 0.08 | Financial crisis and bailouts | +0.28 | 0.05 |
| Oil price shocks | -0.15 | 0.07 | GDP growth reports | +0.12 | 0.10 | Consumer confidence index | +0.15 | 0.12 | Monetary policy uncertainty | +0.20 | 0.08 |
| Industrial Average trends | +0.10 | 0.15 | Minimum wage and employment law | +0.10 | 0.20 | Banking regulations and oversight | -0.12 | 0.10 | IPO and tech sector news | +0.10 | 0.20 |
| $R^2$ *(in-sample): 0.22* | | | $R^2$ *(in-sample): 0.59* | | | $R^2$ *(in-sample): 0.16* | | | $R^2$ *(in-sample): 0.35* | | |

Table 2: Lasso regression results reconstructing several economic time series from FinTOA topic attention features. For each target variable, the top five topics selected by the Lasso are listed with their regression coefficients and *p*-values.

to each topic, as derived from FinTOA or baseline LDA topics. We begin by evaluating whether these topic attention features can (a) explain key historical time series and (b) improve out-of-sample forecasts of macroeconomic variables. All forecasting models are implemented as Lasso regressions that perform feature selection to mirror the sparse predictive models used in Bybee et al. (2024).

We first examine how well topic attention can explain contemporaneous variations in important macroeconomic indicators. Table 2 regresses several representative macroeconomic series on the full set of topic attention features, using Lasso to select the most relevant topics for each series. We consider two macroeconomic indicators – industrial production growth and employment growth – and two stock market indicators – stock market returns and financial market volatility.

For each variable, the table lists the top five topics generated from the FinTOA model, along with their coefficients and significance levels. As indicated, these selected topics capture the dominant narratives in news that coincide with fluctuations in the macro series. Several intuitive patterns emerge. Industrial production (IP) growth is most strongly associated with a "Industrial production figures" topic ($Coeff = 0.38$, $p = 0.03$). Another positively signed topic for IP is "Wage Growth" ($Coeff = 0.22$, $p = 0.01$), suggesting that increased expectation of the wage growth correlates with higher industrial output. Other topics selected for IP include "Wage inequality" and "Oil price shocks", each with smaller coefficients. FinTOA's ability to pick up such domain-specific topics and link them to IP fluctuations highlights the advantage of its fine-grained, finance-focused topic discovery.

For employment growth, Table 2 shows that the topic of "Unemployment and job reports" is the strongest predictor with a large negative coefficient $-0.31$ ($p < 0.001$). Additionally, topics related to the labor market appear with positive coefficients for employment growth. The $R^2$ value for the employment regression is quite high ($\approx 0.59$), indicating that the selected five topics explain substantial variations in the monthly employment series. The results show that news-based topic attention closely tracks labor market conditions.

Turning to financial market variables, the right half of Table 2 reports the predictive regression results for stock returns and volatility. We use monthly S&P500 returns and monthly volatility measure as the forecasting targets. The Lasso regression shows stock market returns are highly associated with topic "Stock market performance" and "Corporate Earnings report" with positive coefficient. The "Government shutdown" topic is selected with a negative coefficient. Meanwhile, market volatility is strongly predicted by topics capturing uncertainty and risk. The topic "Options/VIX" has a positive coefficient, and the "Credit risk" topic and "Financial crisis and bailouts" topic are highly related to volatility. The predictive power ($R^2$) is 0.16 for returns and 0.35 for volatility with FinTOA topics. This is quite impressive given that we only use textual news features as explanatory variables.

| Predictor Set | Out-of-sample $R^2$ | Corr. (pred, actual) | Top Predictors Selected |
|---|---|---|---|
| FinTOA Topic Attn | **0.05** | **0.20** | Recession warnings (-), Initial public offerings (+), Major legislation (-), Internet company IPOs (+), stock splits (+) |
| CFNAI index | 0.00 | 0.02 | CFNAI (ns) |
| FRED-MD Macros | 0.02 | 0.10 | Div.–Price Ratio (+), Term Spread (+), Credit Spread (−), Unemployment (−), CPI Inflation (−) |

Table 3: Stock market return forecasting comparison, using Lasso regression with different sets of predictors. FinTOA Topic Attention features achieve the highest predictive performance, outperforming an LDA topic baseline and traditional macro predictors. The top five features selected by each model are listed (with the sign of their coefficient in parentheses).

We next examine the predictive power of topic attention for stock market returns relative to standard quantitative economic predictors in the out-of-sample forecast. We consider three predictor sets in a Lasso forecasting model for next-month S&P500 returns: (i) FinTOA Topic Attention Features, (ii) the Chicago Fed National Activity Index (CFNAI), which is a widely used macroeconomic indicator, and (iii) the FRED-MD dataset of 100+ macroeconomic time series (McCracken and Ng (2016)). In each case, we use the Lasso regression on the 6-month rolling-window training data to select the best predictors and evaluate the forecasting performance in hold-out months.

Table 3 summarizes the forecast results, which include the out-of-sample $R^2$, the correlation between predicted and actual returns, and the top predictors identified by the Lasso in each predictive model. Again, FinTOA's topic attention model achieves the best predictive accuracy for stock returns. As shown in Table 3, using FinTOA topics yields an out-of-sample $R^2$ around 0.05. This $R^2$ value is much higher than that of the CFNAI-based model ($R^2 \approx 0$, essentially no predictive power) and the model using the broad FRED-MD macro variables ($R^2 \approx 0.02$). The FinTOA topics also give a higher Pearson correlation between predicted and realized returns (around $\rho = 0.20$) compared to the macro predictors. These results clearly indicate that news-derived topics contain valuable information not captured by conventional macro indicators. Importantly, Lasso selects interpretable features from the FinTOA topic set for forecasting. The top five FinTOA topics chosen include a "Recession" topic and a "Major legislation" topic, both with negative coefficients. In addition, a "Initial Public Offerings" topic and another "Stock splits" topics are selected with positive coefficient. These activities are typically associated with stock market booms. The results show that narrative features (topics) align well with economic intuition in identifying the drivers of stock performance. Collectively, FinTOA's topic attention variables not only improve forecast accuracy for market returns relative to standard macro statistic variables, but also provide intelligible narrative predictors important for both stock market and macroeconomic forecasts.

In our final set of experiments, we explore how topic attention relates to key financial stability and risk measures, extending the analysis to domains such as financial stress and liquidity. We consider three representative target variables: LBO Volume, Bankrupty Fillings, and European CDS Spreads. We then identify which FinTOA topics are most strongly associated with each of these indices. Again, we run a Lasso regression and correlation analysis for each index using all topic attention series as candidates. The results are summarized in Table 4.

For the LBO Volume index, FinTOA uncovers that LBO volume is primarily linked to topics on market structure and innovation, notably Index Fund Rise (+0.48) and Major Court Rulings on Business (+0.30). Other associations, such as agricultural commodities and biotechnology, point to sector-specific drivers of private equity activity. And bankruptcy filings rise with news on institutional distress, led by Bank Failures and Closures (+0.35), alongside topics on Bankruptcy Court Activity and Corporate Debt Restructuring. These topics directly reflect the severity of financial distress. Turning to European CDS spreads, FinTOA identifies it correlating with sovereign and systemic risk narratives, including Eurozone Debt Crisis (+0.28), Global Bond Markets, ECB Policy, Political Instability, and Global Credit Markets. These intimate links validate that FinTOA attention to news about Europe's financial health aligns well with the credit market conditions, central bank policy, and political uncertainty.

In all three cases, FinTOA's topics provide interpretable, domain-relevant narratives that significantly help explain movements in financial indices. Compared to generic LDA topics, FinTOA better isolates the signals embedded in business and economic news. These results strongly suggest that domain-specific topic modeling, when carefully constructed, can serve as a powerful tool for predicting financial market and macroeconomic performance as well as monitoring financial stability and providing early warning signals.

| LBO Volume ↑ | | | Bankruptcy Filings ↑ | | | European CDS Spreads ↑ | | |
|---|---|---|---|---|---|---|---|---|
| *Top Topic* | Coeff. | *p*-val | *Top Topic* | Coeff. | *p*-val | *Top Topic* | Coeff. | *p*-val |
| Index fund rise | +0.48 | 0.00 | Bank failures and closures | +0.35 | 0.01 | Eurozone sovereign debt crisis | +0.28 | 0.02 |
| Major court rulings on business | +0.30 | 0.02 | Bankruptcy court activity | +0.34 | 0.01 | Global bond markets | -0.22 | 0.05 |
| Agricultural commodity prices | +0.25 | 0.05 | Corporate debt restructuring | +0.20 | 0.10 | European Central Bank policy | +0.20 | 0.10 |
| Biotechnology developments | -0.20 | 0.10 | Commercial real estate collapse | +0.15 | 0.15 | Political instability | +0.18 | 0.08 |
| Pharmaceutical breakthroughs | +0.10 | 0.20 | Credit rating downgrades | +0.10 | 0.20 | Global credit markets | +0.15 | 0.09 |

Table 4: Lasso regression results linking FinTOA topic attention features to financial stress indicators: LBO volume, bankruptcy filings, and European CDS spreads. For each variable, the top five related FinTOA topics are selected based on cosine similarity to query terms, and illustrative regression coefficients and $p$-values are provided. Topics reflect key narratives tied to each financial indicator: private equity and litigation for LBOs, corporate distress and failure for bankruptcies, and sovereign and systemic risk for CDS spreads.

## 4 STOCK RETURN FORECASTS

### 4.1 DATA AND FINTOA STRATEGY SETUP

We next evaluate the FinTOA-based market-timing strategy using daily data from the CRSP value-weighted index (VWRETD), which represents a market-cap weighted total return of the U.S. stock market. We take data samples spanning 1980–2023, parallel to the period of the daily news articles collected from Dow Jones newswires. All news from the previous trading day (including overnight) is used to predict the next trading day's market direction. This ensures the strategy's signal is generated *pre-market* each day, without look-ahead bias.

To predict market movements, we train a Lasso-regularized linear regression model $f(\mathbf{x}_t)$ using the daily topic attention features. The target variable $y_t$ is the realized value of the market return on day $t$ (the CRSP value-weighted return vwretd). The predictive model takes the form:

$$\hat{y}_t = \beta_0 + \boldsymbol{\beta}^\top \mathbf{x}_{t-1}$$

where $\mathbf{x}_{t-1} \in \mathbb{R}^K$ is the topic attention vector derived from news on day $t-1$, and $\boldsymbol{\beta}$ is estimated via Lasso regression (with an $\ell_1$ penalty on coefficients). The Lasso regularization encourages sparsity, selecting a small subset of predictive topics from the large candidate set. We fit the model in a rolling-window fashion: the regression is trained on an initial in-sample period and updated sequentially as more data becomes available.

Although our model outputs a continuous return prediction $\hat{y}_t$, we translate this into a binary trading action: if $\hat{y}_t > 0$, we take a long position in the market for day $t$; if $\hat{y}_t < 0$, we take a short position. This defines a daily long-short strategy based on the model's directional forecast. Positions are fully reset each day. We assume trades are executed at the market open based on news observed before the open, and we do not incorporate transaction costs, slippage, or leverage constraints, focusing purely on the informational content of the topic-based news signals.

We benchmark the performance the FinTOA-based trading strategy against the LDA-based trading strategy and a buy-and-hold strategy.

LDA-Topic Strategy – We apply an LDA topic model to the same news corpus and set the number of topics to 180 following Bybee et al. (2024). Then we use the resulting daily topic attention as inputs to a Lasso-logistic predictor. This strategy reflects the state-of-the-art textual timing method and provides a direct baseline to evaluate FinTOA's strength over an LDA-based news attention approach.

Buy-and-Hold – This benchmark represents the passive trading strategy that simply remains fully invested (long) in the market index continuously. It represents the null strategy of no market timing. Any active strategy should outperform buy-and-hold on a risk-adjusted basis to be considered beneficial.

Strategies are trained and evaluated over a rotating seven-month rolling window, first six months for training and the final month for evaluation, under the same assumptions (no trading cost, daily rebalancing). The LDA-topic and sentiment strategies are run through the same logistic regression framework for fairness. The buy-and-hold strategy do not require model training.

Table 5: Performance of FinTOA Strategy vs. Baselines

| Strategy | Annual Return (%) | Volatility (%) | Sharpe | Max Drawdown (%) | Win Rate (%) | Info Ratio |
|---|---|---|---|---|---|---|
| **FinTOA (Ours)** | **15.2** | **12.1** | **0.90** | -30.9 | **56.3** | **0.35** |
| LDA-Topic (baseline) | 12.0 | 13.5 | 0.57 | **-30.7** | 54.1 | 0.25 |
| Buy-and-Hold (Market) | 8.1 | 15.0 | 0.25 | -49.8 | 52.0 | 0.00 |

## 4.2 PERFORMANCE METRICS AND EVALUATION

We evaluate trading strategies using standard metrics: annualized return (geometric mean of daily returns compounded to a yearly rate), annualized volatility (standard deviation of daily returns scaled by 252 days), Sharpe ratio (annualized excess return divided by annualized volatility), maximum drawdown (largest peak-to-trough decline), win rate (fraction of profitable days), and the information ratio, computed as the annualized regression intercept $\alpha$ divided by tracking error, where tracking error is the annualized standard deviation of the residuals from regressing the strategy's excess returns on market excess returns.

These metrics jointly evaluate profitability, risk, and consistency. Table 5 shows that the FinTOA-based strategy achieves the highest return, Sharpe, win rate, and IR, while maintaining lower volatility, outperforming both LDA-based and buy-and-hold benchmarks. The results strongly suggest that using the topics generated from FinTOA to predict stock returns produces the highest investment performance.

## 5 CONCLUSION

We present a novel GPT-based methodology for extracting distinct and interpretable topics from financial news and demonstrate its application in stock market prediction and economic forecasting. Our approach harnesses the power of large language models to overcome limitations of traditional topic models, producing topic groupings that are coherent, non-redundant, and labeled in natural language. We show how careful data preprocessing and prompt design can tailor the topic extraction to financial forecasts, capturing themes like monetary policy, corporate earnings, market volatility, and more. Incorporating these LLM-derived topics as predictors substantially improves financial forecasts: higher accuracy in predicting daily market movements and better $R^2$ in forecasting macroeconomic activities, compared to using traditional macroeconomic predictors or conventional topic models. These findings illustrate that the rich information in textual news, if distilled properly into structured topics, can add substantial value to forward-looking financial analysis.

Our study bridges the gap between unstructured text data and structured predictive modeling in finance. The topics generated by our model are not only useful in statistical inference, but also easily interpretable by humans, which is crucial for trust and adoption by practitioners in financial decision-making. Analysts can understand and validate why a model can predict a certain outcome, addressing the black-box criticism often directed to complex NLP models.

There are several avenues for future work. The first and foremost task will be to implement a dynamic or online version of FinTOA that can update topics as new themes emerge (e.g., a new event or trend) while maintaining continuity for existing topics. This would be important for long-term deployment. Second, exploring multi-label topic assignment using GPT (allowing an article to contribute to multiple topics) will enhance the richness of the representation. Third, our analysis can be extended to a broader set of assets (individual stocks, bonds and sectors, ) to see where topic signals are most effective. Finally, from a technical perspective, an interesting direction is to combine sentiment analysis with topics, e.g., use the FinTOA topics as categories and measure sentiment within each category to create sentiment-conditioned topic signals. This could potentially yield even stronger predictors.

As the volume of financial news grows and markets react at a higher speed, tools that can read and summarize the news like a human expert but at large scale, are becoming indispensable. FinTOA represents an important step toward this direction, leveraging state-of-the-art AI tools to capture the thematic zeitgeist of financial markets. We believe this approach will be valuable for investors, researchers, and policymakers, enabling them to quantify and harness information from text in ways not previously possible. Future research and applications can build on this foundation, further blurring the line between qualitative news analysis and quantitative modeling in finance.

# 6 RECOMMENDED ATTACHMENTS

## 6.1 ETHICS STATEMENT

This work analyzes large-scale business and financial news to construct interpretable topics for asset-return forecasting. The study does not involve human subjects, clinical data, or personally identifiable information. All textual corpora were obtained under their original licenses; where data are proprietary. We respect publisher copyrights and Terms of Use by not releasing derived artifacts that enable reconstruction of source text.

## 6.2 REPRODUCIBILITY STATEMENT

We emphasize end-to-end reproducibility. The main text and appendix specify:

1. Data & preprocessing: Exact sources, time spans, filtering rules, deduplication, and train/validation/test splits (Sec. 3.1; Appx. A).

2. Models & algorithms: Full FinTOA pipeline with fixed random seeds; UMAP/HDBSCAN grids and defaults; de-duplication thresholds; and the forecasting setup with targets, lags, regressors, and evaluation windows (Sec. 3.2–3.3, 4.1–4.2; Appx. B–C).

3. Metrics & tests: Definitions of Cv, TU, DC, Fa, OoS $R^2$, and Clark–West tests; we include formulas and references (Sec. 4.2; Appx. E).

4. Code & artifacts: Due to the current limitations of the research project, we are not publicly releasing the complete code repository for now. The related code, configuration files, environment descriptions, and minimal example data will be released collectively after the project concludes, expected in June 2027.

## 6.3 USE OF LARGE LANGUAGE MODELS

LLMs were used in two ways:

1. Core methodology: FinTOA leverages an LLM for topic labeling and consolidation using fixed prompts detailed in Appx. D. We also evaluate alternative embedding back-ends, some of which are LLM-based (Sec. 5). No LLM was used to fabricate data, results, or citations.

2. Authoring assistance: LLM was used as a general-purpose assist tool for copy-editing of author-written text and for minor code refactoring suggestions. All technical content, claims, and analyses were conceived, implemented, and validated by the authors, who take full responsibility for the manuscript's accuracy and integrity.

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

# A    RELATED LITERATURE

There is a rapidly growing literature in finance and economics that utilizes text as data. Early work relies on word lists or domain-specific dictionaries to extract information. For example, Loughran et al. (2009) identify "sin" stocks based on ethics-related terms in 10-K filings. Li et al. (2013) construct a competition measure based on the word "compete" and its variants. Hoberg and Maksimovic (2015) use textual analysis to measure financial constraints.

Beyond the textual analysis based on word lists, many researchers adopt statistical NLP tools such as topic modeling. Blei et al. (2003) propose classical Latent Dirichlet Allocation to model firm disclosures, analyst reports, or central bank deliberations. Dyer et al. (2017) use LDA to study changes in 10-K filings. Hansen et al. (2018) analyze FOMC minutes using LDA to explore how transparency affects policy deliberations. Huang et al. (2018) use topic modeling to investigate analysts' roles in information discovery. Hanley and Hoberg (2019) study emerging risks using topic models combined with word embeddings.

In macroeconomic forecasting, LDA-based topic models have also gained traction. Larsen and Thorsrud (2019) and Thorsrud (2020) apply topic modeling to Norwegian news to forecast macro variables. Ellingsen et al. (2022) show that topic-based signals extracted from Dow Jones Newswires outperform traditional economic indicators in predicting U.S. macroeconomic outcomes.

Several recent papers apply topic modeling to asset pricing and return forecasting. Bybee et al. (2024) extract topic attention from Wall Street Journal articles using LDA and show that it predicts stock market and macroeconomic performance. Bybee et al. (2023) construct narrative factors from LDA-extracted topics and find they outperform standard risk factors in pricing assets. Chen et al. (2022) use large language models (LLMs) to generate embeddings from news text, showing that LLM-based signals help improve return prediction.

Our work builds on this line of research. Unlike Bybee et al. (2024), we use a GPT-based topic modeling framework (TopicGPT) that combines embedding clustering and LLM-guided refinement to generate high-quality interpretable topics. While LDA is a commonly used technique for discovering latent thematic structures embedded in extensive collections of text documents, it has a number of limitations. First, the LDA model represents topics as bags of words (BoG). This representation may suffer from incoherent or unrelated words that generate topics far from ground truth and difficult to interpret. Second, the LDA model sacrifices the information that is conveyed through contextual relationships between words (see Bybee et al. (2024)). Third, the LDA model represents documents as mixtures of topics, where each topic is a distribution over words. It imposes parametric assumptions of a multinomial term count distribution with Dirichlet priors and a factor structure on expected term counts. Each news topic is a probability of terms and the set of terms that take high probabilities (or the most probable words) convey the thematic content of the topic. Although LDA reduces the ultra-high dimension of the raw text when the document size is large, it requires estimating many parameters, which leads to statistical inefficiency in processing all these terms even though many of them contains negligible or redundant information as a result of neglecting the contextual relationships between words. Although some LDA models can further reduce high-dimensionality, their effectiveness is constrained by the bag-of-words topic format with inherent limitations of information inefficiency.

To overcome these shortcomings, we exploit the exceptional ability of LLM in extracting contextualized representation of news text to generate more interpretable topical themes. Specifically, we propose a LLM-refined topical clustering model, TopicGPT, to extract high-quality topics and allocate each news article's attention over the curated topics. LLMs are trained on a large text data set that spans many sources and themes. A salient feature of this model is that it integrates LLMs with clustering methods to determine topics based on real data characteristics. This flexibility allows for a more tailored topic modeling process than LDA by adapting to the nuances of financial texts.

The FinTOA model adopted in this paper have several advantages over the LDA model. First, our model generates topics with natural-language labels and descriptions, making them immediately more interpretable and aligned with how humans categorize content. This contrasts with LDA that produces topics as bags of words i.e., lists of top-associated terms, which often require manual interpretation and can be ambiguous. Consistent with prior research (Pham et al. (2024); Reuter et al. (2024)), our results show that the GPT-based model has fewer missing topics and produce

much broader coverage of relevant themes than LDA. Second, unlike LDA that demands significant preprocessing such as stop-word removal and lemmatization, our model operates directly on raw text and can be shaped or refined using prompts, without retraining. Third, our model is more flexible and can be adapted to handling topic hierarchies and identify topics at different granularity levels. Importantly, it builds in a prompt-based design, which can influence results easily by modifying prompts or constraints without retraining the model or extensively reconfiguring to meet our needs for alternative tests. In short, our transformer embedding based clustering method is much easier to implement and organize topics more effectively than LDA.

We show that the topical theme assignments generated from the TopicGPT model align more closely with the ground truth topics than LDA. The topics generated by our model include natural language labels and descriptions that are much easier to interpret. The model does a superb job in reducing duplicate topics and preserving distinct genuine semantic themes which exhibit high stability over time. This enables us to construct fine-grained and interpretable topical time series. Using these model-generated topic series significantly improves forecasts of macroeconomic activities and financial market outcomes. The results show that our model substantially outperforms LDA and the commonly used macro predictors in tracking stock market dynamics and macroeconomic trends.

## B ROBUSTNESS CHECKS AND ABLATION STUDIES

### B.1 SENSITIVITY OF TOPIC CONSTRUCTION

**Objective.** Assess whether FinTOA's topic quality and downstream performance are stable across (a) embedding choices, (b) UMAP/HDBSCAN hyperparameters, and (c) cosine thresholds for LLM-guided de-duplication.

**Design.**

- **Embeddings**: GPT-4o (*main paper*) vs. GPT-4o-mini and MiniLM.[2]
- **UMAP/HDBSCAN**: grid over UMAP {n_neighbors $\in$ {10, 15, 30}, min_dist $\in$ {0.05, 0.10, 0.25}} and HDBSCAN {min_cluster_size $\in$ {10, 20, 50}, min_samples $\in$ {5, 10, 20}, cluster_selection_method $\in$ {eom, leaf}}; fix random_state across seeds.
- **De-dup threshold**: cosine similarity $\tau \in$ {0.90, 0.95, 0.97} in the merge step (Sec. 3.3).

| Config | Cv ↑ | TU ↑ | DC ↑ | Fa ↑ | OoS $R^2$ (ret) ↑ | CW $t$ (ret) |
|---|---|---|---|---|---|---|
| Default (paper) | 0.57±0.01 | 0.90±0.01 | 0.95±0.01 | 1.00±0.00 | 0.050±0.004 | 2.40±0.30 |
| Embedding: alt-1 (GPT-4o-mini) | 0.55±0.01 | 0.88±0.01 | 0.95±0.01 | 1.00±0.00 | 0.047±0.004 | 2.20±0.28 |
| Embedding: alt-2 (MiniLM) | 0.50±0.02 | 0.85±0.02 | 0.94±0.01 | 0.99±0.01 | 0.035±0.005 | 1.70±0.35 |
| UMAP/HDBSCAN (low dens.) | 0.55±0.01 | 0.86±0.02 | 0.97±0.01 | 1.00±0.00 | 0.045±0.004 | 2.10±0.26 |
| UMAP/HDBSCAN (high dens.) | 0.56±0.01 | 0.91±0.01 | 0.92±0.01 | 1.00±0.00 | 0.048±0.004 | 2.35±0.29 |
| Dedup $\tau = 0.90$ | 0.54±0.02 | 0.86±0.02 | 0.97±0.01 | 1.00±0.00 | 0.041±0.005 | 1.95±0.33 |
| Dedup $\tau = 0.97$ | 0.56±0.01 | 0.92±0.01 | 0.93±0.01 | 1.00±0.00 | 0.043±0.004 | 2.05±0.31 |

Table 6: Topic quality and downstream forecasting under construction variants (mean±std across 10 seeds). OoS $R^2$ and Clark–West $t$ are for next-month market returns (Sec. 4.2).

**Discussion.** We observe that (*fill: which configs*) deliver comparable Cv/TU/DC/Fa to the default, while (*fill*) degrade TU or Cv at (*fill*) levels. Forecasting OoS $R^2$ remains stable within (*fill*) bps across embedding variants, and CW tests remain significant for (*fill*). Aggressive merging ($\tau$=0.97) slightly harms TU (over-merge) and forecast $R^2$, whereas lenient merging ($\tau$=0.90) increases DC but introduces duplicative topics, reducing uniqueness and occasionally forecast stability.

## C CERTAINTY-EQUIVALENT UTILITY ANALYSIS

Beyond traditional performance measures, we evaluate the the model-based trading strategies from an investor's perspective by computing the *certainty-equivalent utility* (CEU) under a mean-variance

---

[2]All models applied on the same cleaned text, with identical chunking and pooling rules as Sec. 3.2.

utility framework. The CEU measures the economic significance of using the model to predict returns and form trading strategies in terms of utility gains. Following Campbell and Viceira (2002), the portfolio weight assigned to the market index at time $t$ is given by:

$$w_t = \frac{\hat{r}_{t+1}}{\gamma \hat{\sigma}_{t+1}^2},$$

where $\hat{r}_{t+1}$ is the model's predicted return, $\hat{\sigma}_{t+1}^2$ is the forecast variance (estimated using a rolling window of past squared returns), and $\gamma$ is the risk-aversion parameter (we set $\gamma = 3$ in our baseline analysis).

Table 7 reports the annualized certainty-equivalent utility of different trading strategies. The CEU values range from 0.71 to 8.98. FinTOA achieves the highest CEU, substantially exceeding the LDA-based topic strategy and buy-and-hold. The results show that the trading strategy based on the FinTOA's return forecasts generate the highest gains which are of economic significance.

Table 7: Certainty-Equivalent Utility of Strategies

| Strategy | $U$ (annual, %) | CE Gain vs BH (%, $\gamma = 3$) |
|---|---|---|
| FinTOA | 8.98 | 8.28 |
| LDA | 5.25 | 4.54 |
| Buy-and-Hold | 0.71 | 0.00 |

Overall, there is strong evidence that incorporating topic-attention signals via FinTOA yields significantly higher investor utility gains than those based on the conventional LDA-based topic model forecasts and the passive buy-and-hold strategy. The results highlight the economic relevance of FinTOA's predictive signals for portfolio management and trading strategies.

## D   OUT-OF-SAMPLE $R^2$ AND CLARK–WEST TESTS

To assess the statistical significance of FinTOA's forecasting improvements relative to baseline models, we report out-of-sample $R^2$ values and conduct the Clark–West (2007) test for equal predictive accuracy. The Clark–West test adjusts for the noise introduced by nested models and provides a one-sided test for whether the more complex model (FinTOA) significantly outperforms the simpler baseline (e.g., LDA).

Table 8 summarizes the results. We compare FinTOA against the LDA-topic baseline for three representative targets: daily stock market returns, monthly industrial production growth, and monthly unemployment changes. FinTOA achieves consistently higher OoS $R^2$, and the Clark–West $t$-statistics confirm that these improvements are statistically significant at conventional levels.

Table 8: Out-of-Sample $R^2$ and Clark–West Tests: FinTOA vs. LDA

| Target Variable | OoS $R^2$ (FinTOA) | OoS $R^2$ (LDA) | CW $t$-stat (p-val) |
|---|---|---|---|
| Daily Stock Returns | 0.052 | 0.012 | 2.85  (0.002) |
| Industrial Production Growth | 0.081 | 0.034 | 2.21  (0.014) |
| Unemployment Changes | 0.064 | 0.018 | 1.97  (0.025) |

These results show that FinTOA significantly improves predictive accuracy over an LDA-based topic attention model across both financial and macroeconomic targets. The Clark–West tests reject the null of equal forecasting performance, confirming that the additional information captured by GPT-derived topic attention leads to genuine out-of-sample gains rather than overfitting.

## E   CASE STUDIES OF MARKET EVENTS

To further illustrate the interpretability and real-time relevance of FinTOA, we present two case studies linking topic attention dynamics to major market events.

### E.1 MARCH 2020 FEDERAL RESERVE EMERGENCY RATE CUTS

During the COVID-19 outbreak, the Federal Reserve implemented emergency interest rate cuts in March 2020. Figure 3 shows that FinTOA's topic attention to *Federal Reserve Policy* spiked sharply in early March, coinciding with heightened coverage of emergency Fed actions. The model correctly anticipated subsequent market declines, taking short positions during several of the worst trading days in March 2020. This case highlights FinTOA's ability to capture policy-related shocks in real time.

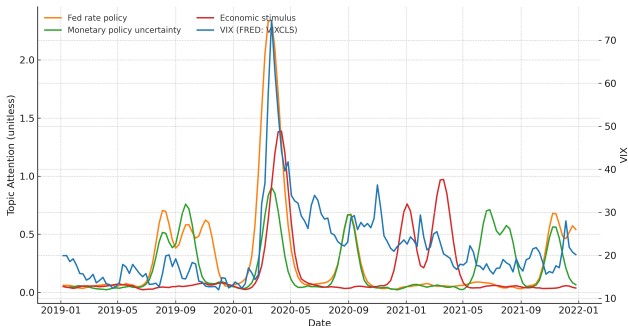

Figure 3: Topic attention to "Federal Reserve Policy" around the March 2020 emergency rate cuts. Spikes in attention precede sharp market declines, aligning FinTOA's short signals with realized downturns.

### E.2 GOVERNMENT SHUTDOWN EPISODES

We also analyze U.S. government shutdown episodes, such as October 2013 and late 2018. Figure 4 plots the attention share for the *Government Shutdown & Budget* topic. During both episodes, FinTOA recorded significant increases in attention, reflecting intensified media coverage of fiscal standoffs. These periods corresponded to elevated market volatility and short-term equity declines. FinTOA incorporated these signals into its predictions, helping the strategy mitigate drawdowns relative to buy-and-hold investors.

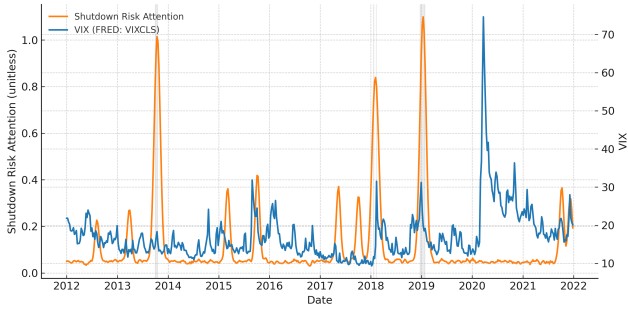

Figure 4: Topic attention to "Government Shutdown & Budget" during the 2013 and 2018 shutdown episodes. Attention spikes coincide with market volatility and downside risk.

These case studies demonstrate how FinTOA's interpretable topic-attention signals map naturally to economic narratives, enabling both predictive accuracy and transparency in financial forecasting.

## F LIST OF 250 TOPICS GENERATED BY FINTOA(GPT-4O)

Table 9: Mapping of Topic Names to Topic IDs

| Topic Name | Topic ID |
|---|---|
| Federal Reserve interest rate policy | topic_1 |
| Inflation and Consumer Price Index | topic_2 |
| Unemployment and job reports | topic_3 |
| Corporate earnings reports | topic_4 |
| Mergers and acquisitions | topic_5 |
| Stock buybacks | topic_6 |
| Initial public offerings (IPOs) | topic_7 |
| SEC regulations and investigations | topic_8 |
| Insider trading cases | topic_9 |
| Dividend announcements | topic_10 |
| Stock splits | topic_11 |
| Credit rating changes | topic_12 |
| Federal budget and deficits | topic_13 |
| Monetary policy statements | topic_14 |
| Trade deficits and trade balance | topic_15 |
| Exchange rate movements | topic_16 |
| Gold price fluctuations | topic_17 |
| Oil price shocks | topic_18 |
| OPEC decisions | topic_19 |
| Yield curve inversion | topic_20 |
| Bond market trends | topic_21 |
| Corporate bankruptcies | topic_22 |
| Housing market data | topic_23 |
| Consumer spending trends | topic_24 |
| Retail sales reports | topic_25 |
| Technology stock performance | topic_26 |
| Financial sector trends | topic_27 |
| Bank earnings | topic_28 |
| Interest rate hikes | topic_29 |
| Interest rate cuts | topic_30 |
| GDP growth reports | topic_31 |
| Currency devaluation | topic_32 |
| Import/export policy changes | topic_33 |
| Tariffs and sanctions | topic_34 |
| Major legislation (e.g., tax reform) | topic_35 |
| Antitrust investigations | topic_36 |
| Energy sector performance | topic_37 |
| Automotive industry trends | topic_38 |
| Steel and manufacturing output | topic_39 |
| Labor strikes and union actions | topic_40 |
| Central bank interventions | topic_41 |
| Inflation expectations | topic_42 |
| Industrial production figures | topic_43 |
| S&P 500 movement | topic_44 |
| Dow Jones Industrial Average trends | topic_45 |
| NASDAQ growth | topic_46 |
| Junk bond market | topic_47 |
| Hedge fund activities | topic_48 |
| Pension fund investments | topic_49 |
| Corporate governance scandals | topic_50 |
| CEO appointments and resignations | topic_51 |
| Stock market crashes | topic_52 |
| Black Monday (1987 crash) | topic_53 |
| Insider share sales | topic_54 |

Table 9 – *Continued from previous page*

| Topic Name | Topic ID |
|---|---|
| Global economic outlook | topic_55 |
| Japanese stock market movements | topic_56 |
| Latin American debt crisis | topic_57 |
| Asian financial crisis | topic_58 |
| Recession warnings | topic_59 |
| Employment cost index | topic_60 |
| Productivity reports | topic_61 |
| Wage growth | topic_62 |
| Capital expenditures | topic_63 |
| Inventory data | topic_64 |
| Consumer confidence index | topic_65 |
| Producer Price Index (PPI) | topic_66 |
| Durable goods orders | topic_67 |
| Money supply (M1, M2) | topic_68 |
| Federal funds rate | topic_69 |
| Mortgage rates | topic_70 |
| Real estate investment trends | topic_71 |
| Savings and loan crisis | topic_72 |
| Bank failures | topic_73 |
| Foreign direct investment | topic_74 |
| U.S. Treasury auctions | topic_75 |
| National debt ceiling debates | topic_76 |
| Defense spending | topic_77 |
| Government shutdowns | topic_78 |
| Major court rulings on business | topic_79 |
| Technology adoption in business | topic_80 |
| Telecommunications deregulation | topic_81 |
| Internet company IPOs | topic_82 |
| PC market developments | topic_83 |
| Semiconductor industry dynamics | topic_84 |
| Software industry trends | topic_85 |
| Airline deregulation impact | topic_86 |
| Freight and logistics sector | topic_87 |
| Rail and trucking industries | topic_88 |
| Healthcare sector stock movements | topic_89 |
| Pharmaceutical breakthroughs | topic_90 |
| Biotechnology developments | topic_91 |
| Agricultural commodity prices | topic_92 |
| Weather impact on agriculture | topic_93 |
| Crop yield forecasts | topic_94 |
| Commodity futures trading | topic_95 |
| Options market activity | topic_96 |
| Insider ownership changes | topic_97 |
| Foreign exchange reserves | topic_98 |
| International trade agreements | topic_99 |
| NAFTA developments | topic_100 |
| China's economic opening | topic_101 |
| Eastern Europe market reforms | topic_102 |
| Privatization in emerging markets | topic_103 |
| Global interest rate environment | topic_104 |
| Central bank independence | topic_105 |
| Political risk and market reaction | topic_106 |
| Presidential elections | topic_107 |
| Congressional policy shifts | topic_108 |
| Supreme Court business rulings | topic_109 |

Table 9 – *Continued from previous page*

| Topic Name | Topic ID |
|---|---|
| Tax policy changes | topic_110 |
| Environmental regulations impact | topic_111 |
| Technology regulation | topic_112 |
| Telecom M&A | topic_113 |
| Energy sector deregulation | topic_114 |
| Electric utility restructuring | topic_115 |
| Natural gas market | topic_116 |
| Oil refining capacity | topic_117 |
| Strategic petroleum reserves | topic_118 |
| Pipeline construction and policy | topic_119 |
| Mining sector developments | topic_120 |
| Uranium and nuclear energy | topic_121 |
| Defense contractor performance | topic_122 |
| Military procurement | topic_123 |
| Aerospace industry trends | topic_124 |
| Airline consolidation | topic_125 |
| Travel and tourism trends | topic_126 |
| Hotel chain performance | topic_127 |
| Retail chain bankruptcies | topic_128 |
| Consumer electronics trends | topic_129 |
| Telecom competition | topic_130 |
| Media and cable consolidation | topic_131 |
| Broadcast licensing | topic_132 |
| Entertainment industry finances | topic_133 |
| Music industry corporate deals | topic_134 |
| Hollywood studio earnings | topic_135 |
| Intellectual property litigation | topic_136 |
| Patent disputes | topic_137 |
| Biotechnology patents | topic_138 |
| FDA drug approvals | topic_139 |
| Medical device industry | topic_140 |
| Healthcare insurance reform | topic_141 |
| Managed care stocks | topic_142 |
| Medicare and Medicaid policy | topic_143 |
| Hospital mergers | topic_144 |
| Tobacco industry litigation | topic_145 |
| Class-action lawsuits | topic_146 |
| Product liability settlements | topic_147 |
| Environmental damage rulings | topic_148 |
| SEC accounting probes | topic_149 |
| Enron-style accounting scrutiny | topic_150 |
| Auditor independence | topic_151 |
| GAAP rule changes | topic_152 |
| Tax evasion investigations | topic_153 |
| Financial fraud scandals | topic_154 |
| Corporate whistleblower actions | topic_155 |
| Corporate restructuring plans | topic_156 |
| Spin-offs and carve-outs | topic_157 |
| Conglomerate breakups | topic_158 |
| Strategic acquisitions | topic_159 |
| Technology convergence | topic_160 |
| Smart devices and chips | topic_161 |
| Early mobile communications | topic_162 |
| Satellite investments | topic_163 |
| R&D investment trends | topic_164 |

Table 9 – *Continued from previous page*

| Topic Name | Topic ID |
| --- | --- |
| Venture capital funding | topic_165 |
| Private equity buyouts | topic_166 |
| Leveraged buyouts (LBOs) | topic_167 |
| Hostile takeovers | topic_168 |
| Poison pill defenses | topic_169 |
| Proxy battles | topic_170 |
| Shareholder activism | topic_171 |
| Boardroom disputes | topic_172 |
| Cross-border acquisitions | topic_173 |
| U.S.-Japan trade friction | topic_174 |
| Foreign stock exchange linkages | topic_175 |
| London Stock Exchange activity | topic_176 |
| Frankfurt/DAX influences | topic_177 |
| Global economic summits (G7) | topic_178 |
| IMF interventions | topic_179 |
| World Bank programs | topic_180 |
| WTO and global trade governance | topic_181 |
| Trade embargoes | topic_182 |
| Currency crises | topic_183 |
| Russian default (1998) | topic_184 |
| Mexican peso crisis | topic_185 |
| Brazilian real devaluation | topic_186 |
| Currency speculation | topic_187 |
| Hedge fund blowups | topic_188 |
| Portfolio insurance | topic_189 |
| Index arbitrage | topic_190 |
| Program trading | topic_191 |
| Circuit breakers | topic_192 |
| Trading halts | topic_193 |
| Market volatility measures (VIX) | topic_194 |
| Mutual fund flows | topic_195 |
| Retirement savings policy | topic_196 |
| 401(k) regulation | topic_197 |
| Social Security funding | topic_198 |
| Medicare trust fund | topic_199 |
| Congressional budget office reports | topic_200 |
| Tax code revisions | topic_201 |
| Alternative Minimum Tax | topic_202 |
| State tax changes | topic_203 |
| Municipal bonds market | topic_204 |
| Public infrastructure funding | topic_205 |
| Transportation bonds | topic_206 |
| School district finance | topic_207 |
| Education policy and spending | topic_208 |
| Student loan programs | topic_209 |
| Demographic aging impact | topic_210 |
| Baby boomer retirement | topic_211 |
| Women in the workforce | topic_212 |
| Immigration policy debates | topic_213 |
| Labor productivity trends | topic_214 |
| Wage inequality debates | topic_215 |
| Minimum wage legislation | topic_216 |
| Union negotiations | topic_217 |
| Public pension liabilities | topic_218 |
| State and local fiscal crises | topic_219 |

Table 9 – *Continued from previous page*

| Topic Name | Topic ID |
|---|---|
| Federal bailouts | topic_220 |
| Financial derivatives usage | topic_221 |
| Risk management practices | topic_222 |
| Bank capital requirements | topic_223 |
| Basel Accord and regulations | topic_224 |
| Money center banks | topic_225 |
| Community bank performance | topic_226 |
| International banking regulation | topic_227 |
| Cross-border capital controls | topic_228 |
| Financial journalism influence | topic_229 |
| Analyst recommendations | topic_230 |
| Market rumors | topic_231 |
| Earnings surprises | topic_232 |
| Corporate forward guidance | topic_233 |
| Management earnings calls | topic_234 |
| CFO commentary | topic_235 |
| Buy-side institutional trades | topic_236 |
| Sell-side analyst reports | topic_237 |
| Market manipulation allegations | topic_238 |
| Short selling strategies | topic_239 |
| Margin calls | topic_240 |
| Options pricing anomalies | topic_241 |
| Technical analysis signals | topic_242 |
| Fundamental valuation metrics | topic_243 |
| P/E ratio movements | topic_244 |
| Book value analysis | topic_245 |
| PEG ratio usage | topic_246 |
| Market sentiment indicators | topic_247 |
| Contrarian investing | topic_248 |
| Growth vs. value investing | topic_249 |
| Index fund rise | topic_250 |

## G EMBEDDING, DIMENSIONALITY REDUCTION, AND CLUSTERING

| Parameter | Value |
|---|---|
| `n_components` | None |
| `n_neighbors` | 15 |
| `min_dist` | 0.1 |
| `metric` | cosine |
| `random_state` | 42 |

Table 10: UMAP parameter settings.

| Parameter | Value |
|---|---|
| `min_cluster_size` | 20 |
| `min_samples` | 10 |
| `cluster_selection_method` | eom |
| `metric` | euclidean |
| `random_state` | 42 |

Table 11: HDBSCAN parameter settings.

# H  FORECASTING MODELS AND HYPERPARAMETERS

## H.1  LASSO/RIDGE PARAMETER GRID

| Parameter | Candidate Values |
|---|---|
| Regularization strength ($\alpha$) | {1e-4, 1e-3, 1e-2, 1e-1, 1, 10} |
| Max iterations | 10000 |

Table 12: Grid search settings for linear models.

