# OpenReview forum: "A LLM-Refined Dynamic Topic Clustering Framework for Business Forecasts with Large Corpora"
_ICLR.cc/2026/Conference — Submitted to ICLR 2026_

### Official Review · Reviewer_jWE8 · 2025-10-29

**Soundness:** 2
**Presentation:** 2
**Contribution:** 2
**Rating:** 2
**Confidence:** 4

**Summary:**

This paper proposes FinTOA, an LLM-refined topic clustering framework for financial topic discovery that combines embeddings, UMAP, HDBSCAN, and GPT-4-based labeling. In the first stage, FinTOA aims to extract a set of high-quality, distinct, and human-readable topics from a large financial news corpus. In the second stage, it quantifies each article’s relevance to the extracted topics by prompting an LLM to assign “attention” scores across all topics. Generally, FinTOA builds on and extends the existing TopicGPT (Reuter et al., 2024), a prompt-based topic model that integrates GPT’s strengths into the topic extraction process. The authors validate the framework through topic quality metrics and downstream applications.

**Strengths:**

- The paper presents a well-structured pipeline that logically combines document embeddings, dimensionality reduction, clustering, and LLM-based refinement.
- FinTOA outperforms several traditional topic modeling baselines，achieves higher prediction accuracy on macroeconomic forecasting.

**Weaknesses:**

- The paper merely applies the existing TopicGPT method to the financial domain without making significant algorithmic contributions, rendering it to fall below the standard of ICLR.
- The authors extensively use LLMs for topic prediction and merging throughout the pipeline but provide neither the prompts nor output examples, making the entire workflow opaque and impossible to reproduce or evaluate.
- Although TopicGPT and BERTopic are mentioned as representative methods, direct experimental comparisons with these baselines on the same prediction tasks are entirely missing.
- Figure 2 is blurry, and the component definitions are unclear; the authors must provide high-quality vector graphics (SVG/PDF) with explicit data flow annotations.
- There are two methods with the same name, TopicGPT, i.e., one proposed by Reuter et al. in 2024 and the other by Pham et al. in 2024. The former was only published on arXiv and the latter was presented at NAACL 2024. It is suggested that the authors detail the differences between the above two methods and explain why they do not compare their method with them.

**Questions:**

- The paper employs two disconnected evaluation frameworks—topic quality metrics in Section 3.1 (Coherence, Uniqueness, Coverage, Factuality) and downstream trading performance in Section 4. Is there a demonstrated positive correlation between Topic Quality Metrics and Downstream Trading Performance?
- Why does the comparison use LDA with 180 topics, BERTopic with unspecified default topics, and FinTOA with 250 topics?

---

### Official Review · Reviewer_nHx7 · 2025-10-30

**Soundness:** 2
**Presentation:** 3
**Contribution:** 2
**Rating:** 2
**Confidence:** 4

**Summary:**

The authors propose a new framework to model financial news topics using large language models (LLMs) combined with document embeddings, clustering, and GPT4-based topic refinement. They have used conventional Topic models such as LDA and BERTopic to comapare with their proposed Framework FinTOA (Financial Topic Attention). FinTOA combines embedding-based clustering (UMAP + HDBSCAN) with GPT-4-assisted topic refinement and attention scoring, in contrast to previous works (e.g., LDA, BERTopic, and TopicGPT). The effectiveness of LLM-generated Topics is assessed quantitatively and qualitatively.

**Strengths:**

1. This paper presents an approach that integrates large language models (LLMs) into topic modeling and financial forecasting.
2. Incorporating LLMs into topic models to merge the semantically similar topics, can enhances topic diversity and overall quality.
3. The authors clearly outline their motivation and methodology in a step-by-step manner, supported by informative figures.
4. The evaluation framework includes quantitative metrics for topic quality, such as coherence, uniqueness, factuality, and coverage, as well as forecasting performance benchmarks based on Lasso Regression.

**Weaknesses:**

1. Eventhough the title says dynamic Topic clustering, the model used for comparision is not dynamic Topic model. Some of the dynamic Topic model that can be used to evaluate Topic over time are DETM (Dynamic Embedded Topic Model), DLDA (Dynamic Latent Dirichlet Allocation), Dynamic BERTopic, DNLDA (Dynamic Noiseless LDA), CFDTM (Chain-Free Dynamic Topic Model).

2. No computational efficiency discussion: FinTOA processes 1.2M documents in 800-doc batches using OpenAI embeddings + GPT-4 labeling. This implies massive API cost and compute time, but the paper provides No runtime, No cost Analysis.

3. The hyperparameter settings for the models are not provided, even though the performance of LDA and BERTopic depends heavily on these configurations.

4. For FinTOA, the paper states (line 057) that “in the second stage, it quantifies each article’s relevance to these topics by prompting an LLM to assign attention scores across the discovered topics.” However, the actual prompt used is not provided. Did the authors conduct experiments with different prompts?

5. More Topic modeling (Contextualized Topic Model, ProdLDA, Neural Variational Correlated Topic Modeling, Embedded Topic Model) can be added to compare FinTOA, right now only LDA, BERTopic is used.

6. The paper does not include a reference or citation for the financial news dataset used.

7. The paper does not specify the number of runs conducted for the experiments.

8. Figure 1 appears undersized, and the text in Tables 2 and 3 is hard to read.

**Questions:**

1. In line 188-190, ' If the cosine similarity between two topics exceeds a threshold of 0.95, we automatically merge these topics into
a single representative topic.' How do you merge the Topic?

2. Why the number of Topic used for LDA is 180? What are the hyperparameters used for Training LDA model?

3. It is unclear how many top words per topic were considered when calculating coherence for FinTOA, LDA, and BERTopic.

4. Since LDA, BERTopic, and FinTOA produce different numbers of topics, how did you make the Cv (coherence) comparison fair? Did you normalize for topic count or evaluate each model’s natural configuration?

---

### Official Review · Reviewer_iYiM · 2025-11-01

**Soundness:** 3
**Presentation:** 2
**Contribution:** 2
**Rating:** 2
**Confidence:** 4

**Summary:**

This paper aims to propose a topic model for financial news articles and event forecasting. Specifically, this paper builds upon prompt-based topic modeling approach where authors use GPT-based LLM to extract and refine topics from corpus. Authors further propose an LLM prompting technique to measure each article’s attention to all the identified topics. Experiments verify the effectiveness of the proposed method.

**Strengths:**

1. This paper uses the trendy LLMs to enhance topic modeling, which is novel and significant. The application of the proposed method in the financial domain is interesting as well. The overall writing of the paper is decent, with figures as visual illustrations.

2. Experiments are conducted on multiple evaluation metrics, which comprehensively verify the effectiveness of the proposed method.

3. Authors further identify some research directions for future study, which opens avenues for follow-up research.

**Weaknesses:**

1. Usually when we do experiments, we encourage authors to repeat the same experimental setting multiple times and report both mean and standard deviation. However, this paper shows mean but not stddev, which is difficult for readers to judge how significantly the proposed method outperforms baselines.

2. For topic modeling, authors are encouraged to provide a visualization for the learned document representations to visually and intuitively understand what the model learns. This paper lacks such a visualization task.

3. This paper mainly compares to LDA and BERTopic as baselines, which are insufficient. Authors are suggested to compare to more recent baselines, such as [1].

[1] Wu, X., Nguyen, T., Zhang, D., Wang, W. Y., & Luu, A. T. (2024). Fastopic: Pretrained transformer is a fast, adaptive, stable, and transferable topic model. Advances in Neural Information Processing Systems, 37, 84447-84481.

**Questions:**

N/A.

---

### Meta-Review · Area_Chair_xVFR · 2026-01-04

**Summary:**

This work proposes a new topic model that is based on an integration of clustering and LLM.


Strengths: The reviewers acknowledge that using LLM in topic models would be beneficial and that some performance enhancement has been seen in the paper.

Weaknesses: The reviewers are not satisfied by the lack of baselines, especially those related to dynamic topic models. They also question the validity of the experiments as there seems to be no standard deviation reported. A reviewer mentioned that this work relies heavily on TopicGPT without significant algorithmic innovation and also questioned the reproducibility.

**Reviewer Concerns:**

there was no rebuttal

**Reviewer Scores:**

there was no rebuttal

---

### Decision · Program_Chairs · 2026-01-26

Reject